# Quality of Life Longitudinal Evaluation in Prostate Cancer Patients from Radiotherapy Start to 5 Years after IMRT-IGRT

Angelo Maggio [1,*], Tiziana Rancati [2], Marco Gatti [1], Domenico Cante [3], Barbara Avuzzi [2], Cinzia Bianconi [4], Fabio Badenchini [2], Bruno Farina [5], Paolo Ferrari [6], Tommaso Giandini [2], Giuseppe Girelli [5], Valeria Landoni [7], Alessandro Magli [8], Eugenia Moretti [8], Edoardo Petrucci [3], Paolo Salmoiraghi [9], Giuseppe Sanguineti [7], Elisa Villa [9], Justyna Magdalena Waskiewicz [6], Alessia Guarneri [1], Riccardo Valdagni [2,10], Claudio Fiorino [4] and Cesare Cozzarini [4]

1   Istituto di Candiolo-FPO, IRCCS, 10060 Candiolo, Italy; marco.gatti@ircc.it (M.G.); alessia.guarneri@ircc.it (A.G.)
2   Fondazione IRCCS Istituto Nazionale dei Tumori di Milano, 20133 Milano, Italy; tiziana.rancati@istitutotumori.mi.it (T.R.); avuzzi.barbara@istitutotumori.mi.it (B.A.); fabio.badenchini@istitutotumori.mi.it (F.B.); tommaso.giandini@istitutotumori.mi.it (T.G.); riccardo.valdagni@unimi.it (R.V.)
3   Ospedale di Ivrea, A.S.L. TO4, 10015 Ivrea, Italy; dcante@aslto4.piemonte.it (D.C.); epetrucci@aslto4.piemonte.it (E.P.)
4   IRCCS Ospedale San Raffaele, 20132 Milano, Italy; bianconi.cinzia@hsr.it (C.B.); fiorino.claudio@hsr.it (C.F.); cozzarini.cesare@hsr.it (C.C.)
5   Ospedale degli Infermi, 13875 Biella, Italy; bruno.farina@aslbi.piemonte.it (B.F.); giuseppe.girelli@albi.pemont.it (G.G.)
6   Comprensorio Sanitario di Bolzano, 39100 Bolzano, Italy; paolo.ferrari@sabes.it (P.F.); justynamagdalena.waskiewicz@sabes.it (J.M.W.)
7   IRCCS Istituto Tumori Regina Elena, 00144 Roma, Italy; valeria.landoni@ifo.gov.it (V.L.); giuseppe.sanguineti@ifo.it (G.S.)
8   Ospedale di Udine, 33100 Udine, Italy; alessandro.magli@asufc.sanita.fvg.it (A.M.); eugenia.moretti@ausfc.sanita.fvg.it (E.M.)
9   Cliniche Gavazzeni-Humanitas, 24121 Bergamo, Italy; paolo.salmoiraghi@gavazzeni.it (P.S.); elisa.villa@gavazzeni.it (E.V.)
10  Department of Oncology and Hemato-Oncology, Università degli Studi di Milano, 20122 Milano, Italy
*   Correspondence: maggio.angelo@gmail.com

**Abstract:** Purpose: The purpose of this study is to study the evolution of quality of life (QoL) in the first 5 years following Intensity-modulated radiation therapy (IMRT) for prostate cancer (PCa) and to determine possible associations with clinical/treatment data. Material and methods: Patients were enrolled in a prospective multicentre observational trial in 2010-2014 and treated with conventional (74–80 Gy, 1.8–2 Gy/fr) or moderately hypofractionated IMRT (65–75.2 Gy, 2.2–2.7 Gy/fr). QoL was evaluated by means of EORTC QLQ-C30 at baseline, at radiation therapy (RT) end, and every 6 months up to 5 years after IMRT end. Fourteen QoL dimensions were investigated separately. The longitudinal evaluation of QoL was analysed by means of Analysis of variances (ANOVA) for multiple measures. Results: A total of 391 patients with complete sets of questionnaires across 5 years were available. The longitudinal analysis showed a trend toward the significant worsening of QoL at RT end for global health, physical and role functioning, fatigue, appetite loss, diarrhoea, and pain. QoL worsening was recovered within 6 months from RT end, with the only exception being physical functioning. Based on ANOVA, the most impaired time point was RT end. QoL dimension analysis at this time indicated that acute Grade $\geq$ 2 gastrointestinal (GI) toxicity significantly impacted global health, physical and role functioning, fatigue, appetite loss, diarrhoea, and pain. Acute Grade $\geq$ 2 genitourinary (GU) toxicity resulted in lower role functioning and higher pain. Prophylactic lymph-nodal irradiation (WPRT) resulted in significantly lower QoL for global health, fatigue, appetite loss, and diarrhoea; lower pain with the use of neoadjuvant/concomitant hormonal therapy; and lower fatigue with the use of an anti-androgen. Conclusions: In this prospective, longitudinal, observational study, high radiation IMRT doses delivered for PCa led to a temporary worsening of QoL, which tended to be completely resolved at six months. Such transient worsening was mostly associated with acute GI/GU toxicity, WPRT, and higher prescription doses.

**Keywords:** prostate cancer; radiotherapy; EORTCQLQ-C30 questionnaire; quality of life

## 1. Introduction

Radiation therapy is a widely used, highly effective, therapeutic modality for the definitive treatment of locally advanced or localized prostate cancer (PCa), with or without the combination of androgen deprivation [1–6]. With the continuous improvement of clinical outcome, radiation oncologists increasingly pay attention to the possible impact of side effects on an individual's functioning, physical, and/or psychosocial domains. For this reason, patient-reported outcomes (PROs) concerning health-related quality of life (HRQoL) need to be increasingly considered in order to better investigate more subjective parameters (e.g., physical, social, or role functioning), possibly allowing for the identification of bothersome symptoms not highlighted by physician-reported toxicity. Consequently, patients' HRQoL is more and more becoming a noteworthy factor supporting the decisions regarding treatment options [7]. Donovan et al. [8] compared the subjective outcomes of 1643 men managed with active monitoring, radical prostatectomy, or external-beam radiotherapy (EBRT) for their clinically localized prostate cancer. The patient-reported HRQoL evaluated by means of questionnaires emphasized that prostatectomy had the greatest negative effect on the patients' sexual and urinary function while EBRT had only a little effect on urinary continence. Similar findings were reported by systematic reviews focused on a quality of life comparison among surgery, radiotherapy, surveille, or brachytherapy [9]. Additional studies evaluating changes in quality of life in PCa patients undergoing radiotherapy and that aimed at identifying factors that influence QoL were performed. Yucel et al. [10], evaluating 367 PCa patients treated with definitive RT, observed a transient radiation-induced deterioration of patients' HRQoL, with complete restoration by one month from radiotherapy end. Furthermore, a correlation between HRQoL and disease-specific and patient-specific factors was found.

One of the most widely used instruments to assess the QoL of cancer patients is the EORTC QLQ-C30 questionnaire [11] developed in 2001 by the European Organisation for Research and Treatment of Cancer (EORTC).

The aim of the current paper was to investigate patients' HRQoL changes in the first 5 years following intensity-modulated radiotherapy (IMRT) for prostate cancer in a large, prospectively followed multicentric cohort and to determine possible associations with clinical and treatment factors.

## 2. Materials and Methods

Between April 2010 and December 2014, 391 patients treated with definitive IMRT for both clinically localized or locally advanced, non-metastatic, prostate cancer were enrolled in a prospective, longitudinal, multicentre observational trial aimed at developing predictive models of both urinary toxicity and erectile dysfunction. The DUE-01 (Urinary and Erectile Disfunction after radical RT for localized prostate cancer) study was approved by the local Ethical Committee of any participating institution, and written informed consent for the inclusion in the study was obtained for each enrolled patient. The study's coordinator centre Ethics Committee Name is "IL COMITATO ETICO dell'Ospedale San Raffaele–Milano, Istituto di Ricovero e Cura a Carattere Scientifico" with approval protocol N°: DUE-01.

Detailed information about the trial was reported previously [12,13]. In short, relevant clinical, dosimetric, and patient-reported toxicity data were prospectively collected. Patients were treated at different prescription doses with conventional (74–80 Gy, 1.8–2 Gy/fr) or moderately hypofractionated IMRT (65–75.2 Gy, 2.2–2.7 Gy/fr), always with 5 fractions/week. The prescribed doses D were converted into 2 Gy equivalent doses (EQD2), according to the linear quadratic model [14] considering the formula $EQD2 = D(\alpha/\beta + d)/(\alpha/\beta + 2)$, where d is the daily dose and the $\alpha/\beta$ ratio is a measure of the fractionation sensitivity of the cells.

This value was set according to the literature-reported data [14,15] depending on the toxicity endpoint investigated.

All patients were treated supine with an empty rectum and comfortably full bladder.

Patients were treated at eight institutions, as shown in Table 1, while patients' characteristics are indicated in Table 2. Figure 1 reports how many patients completed the questionnaire at different time points.

**Table 1.** Number of patients grouped according to fractionation and institution.

| Institution | Patients | | N |
| | Conventional Fractionation | Moderate Hypofractionation | |
|---|---|---|---|
| 1 | 111 | 21 | 132 |
| 2 | - | 76 | 76 |
| 3 | - | 12 | 12 |
| 4 | 13 | 12 | 25 |
| 5 | 9 | 21 | 30 |
| 6 | 27 | 28 | 55 |
| 7 | - | 32 | 32 |
| 8 | - | 29 | 29 |

**Table 2.** Patient characteristics. Counts (percentage in parenthesis) for categorical variables and medians (range) for continuous variables are reported.

| | |
|---|---|
| Age (y) | 71 (67–74) |
| BMI (kg/m$^2$) | 26 (19-42) |
| PSA (ng/mL) | 6.7 (0.3–277) |
| Gleason score: | |
| <7 | 135 |
| =7 | 186 |
| >7 | 40 |
| n.a. | 30 |
| T stage: | |
| T1 | 217 |
| T2 | 117 |
| T3-4 | 46 |
| TX | 11 |
| Lymph node staging | |
| Nx | 349 |
| N0 | 39 |
| N1 | 3 |
| Diabetes | 63 (16%) |
| Cardiovascular disease | 102 (26%) |
| Hypercholesterolemia | 23 (6%) |
| Urological disease | 23 (6%) |
| Anticoagulants | 27 (7%) |
| Antidepressive | 16 (4%) |
| TURP | 39 (10%) |
| Previous abdominal surgery | 180 (46%) |
| Smoke | 63 (16%) |
| Alcohol | 188 (48%) |
| Hormone therapy before/during RT | 227 (58%) |
| Pelvic irradiation (Yes/No) | 167 (Yes 42.7%)/224 (No 57.3%) |
| Hormone therapy after RT 166 (56%) | 226 (58%) |
| Prescribed dose (Gy) | HYPO (n = 231): 70.2 (54.3–74.2) |
| | CONV (n = 160): 76 (74–83.2) |
| Daily dose (Gy/fr) | HYPO: 2.55 (2.2–3.8) |
| | CONV: 2.0 (1.8–2.0) |
| CTV volume (cc) 51 (34–66) | 52 (11–180) |
| PTV volume (cc) 131 (93–170) | 132 (28–350) |

(BMI = body mass index; TURP = transurethal resection of prostate; CTV = clinical target volume; PTV = planning target volume.)

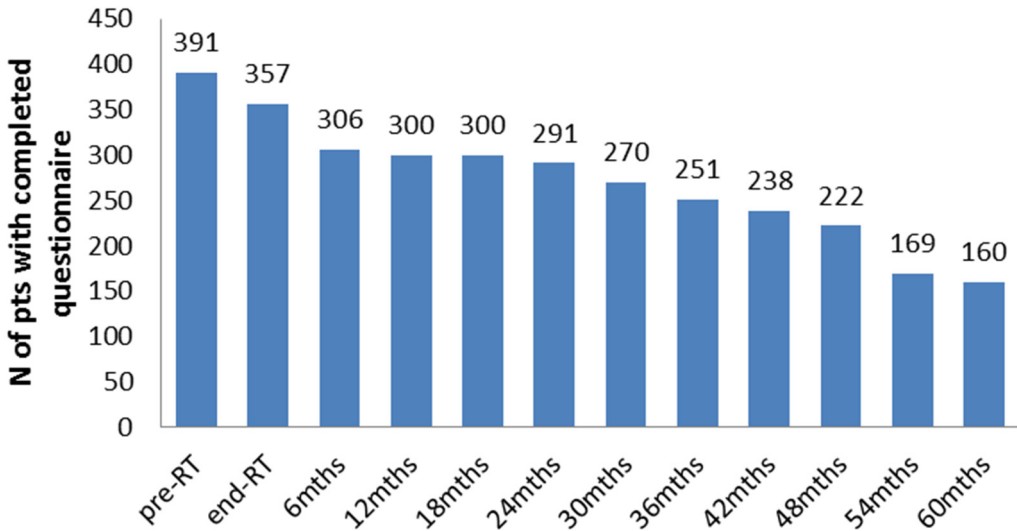

**Figure 1.** Number of patients that completed the questionnaire at different time points.

The health-related quality of life (HRQoL) of prostate cancer patients was assessed by means of the EORTC QoL 30-item questionnaire (EORTC QLQ-C30) filled in by patients at baseline, at RT end, and thereafter every 6 months up to 5 years after IMRT. The 14 domains investigated by the EORTC QLQ-C30 were the following: global health/QoL, five functional scales (physical, role, cognitive, emotional, and social), and eight symptom scales/items (fatigue, nausea/vomiting, pain, dyspnoea, insomnia, appetite loss, constipation, and diarrhoea). Patients' responses were scored according to the EORTC QLQ-C30 scoring manual [11]. With respect to the functional scores, a higher score indicated better functioning levels, whereas a higher score in the symptom scales indicated a higher severity (worse) of symptoms.

The questionnaire scores were longitudinally evaluated across time using repeated-measures analysis of variance for multiple measures (ANOVA). Effects of multiple variables such as age, presence and type of hormonal therapy, prescribed dose, and CTCAE (Common Toxicity Criteria for Adverse Event, v4.03) acute intestinal Grade $\geq 2$ on QoL changes over the time were studied using two-way repeated-measures analysis of variance. Differences among groups were evaluated through the Mann–Whitney test. A *p* value <0.05 was considered to indicate statistical significance.

The statistical analysis was performed using Medcalc statistical software (Version 10; Broekstratt 52, 9030 Mariakerke, Belgium).

### 3. Results

#### 3.1. Longitudinal Changes in QoL Scores

The median age of the patients considered was 71 years. Table 3 reports the summary of the results from the ANOVA analysis while Figure 2 presents longitudinal results for the 14 investigated QoL dimensions. A general trend toward the significant worsening of QoL at RT end with respect to the baseline was detected for global health (5-point worsening, *p* = 0.05), physical (4-point worsening, *p* = 0.04), role functioning (5-point worsening, *p* = 0.04), fatigue (7-point worsening, *p* = 0.03), appetite loss (5-point worsening, *p* = 0.004), diarrhoea (14-point worsening, *p* = 0.05), and pain (5-point worsening, *p* = 0.03). With the only exception of physical functioning which exhibited a further worsening of 4 points at 5 years, all the remaining QoL aspects impaired by RT usually recovered within 6 months from radiotherapy conclusion. No significant variations were, on the contrary, observed for cognitive functioning, insomnia, nausea, dyspnoea, and constipation.

**Table 3.** Results of ANOVA analysis (over 5 years after the end of IMRT) on the 14 QoL dimensions investigated by EORTC QLQ C30 questionnaire.

| Qol Dimension | Significant Trend with Time from ANOVA | *p*-Value | Time and Size of Significant Variation |
|---|---|---|---|
| Global health/QoL | quadratic | 0.05 | 5-point worsening at RT end with respect to baseline, then recovery |
| Physical functioning | cubic | 0.04 | 4-point worsening at RT end with respect to baseline, then recovery, further 4-point worsening at 5 years |
| Role functioning | cubic | 0.04 | 5-point worsening at RT end with respect to baseline, then recovery, further 5-point worsening at 5 years |
| Social functioning | linear | 0.04 | 2-point increase in the 5-year period |
| Emotional functioning | linear | 0.01 | 3-point increase in the 5-year period |
| Cognitive functioning | no significant trend | | |
| Appetite loss | quadratic | 0.004 | 5-point worsening at RT end with respect to baseline, then recovery |
| Diarrhoea | quadratic + linear decrease | 0.05 | 14-point worsening at RT end with respect to baseline, then recovery, and at 5 years, 1.5 points better with respect to baseline |
| Fatigue | quadratic | 0.03 | 7-point worsening at RT end with respect to baseline, then recovery |
| Insomnia | no significant trend | | |
| Dyspnoea | no significant trend | | |
| Pain | quadratic | 0.03 | 5-point worsening at RT end with respect to baseline, then recovery |
| Constipation | no significant trend | | |
| Nausea | no significant trend | | |

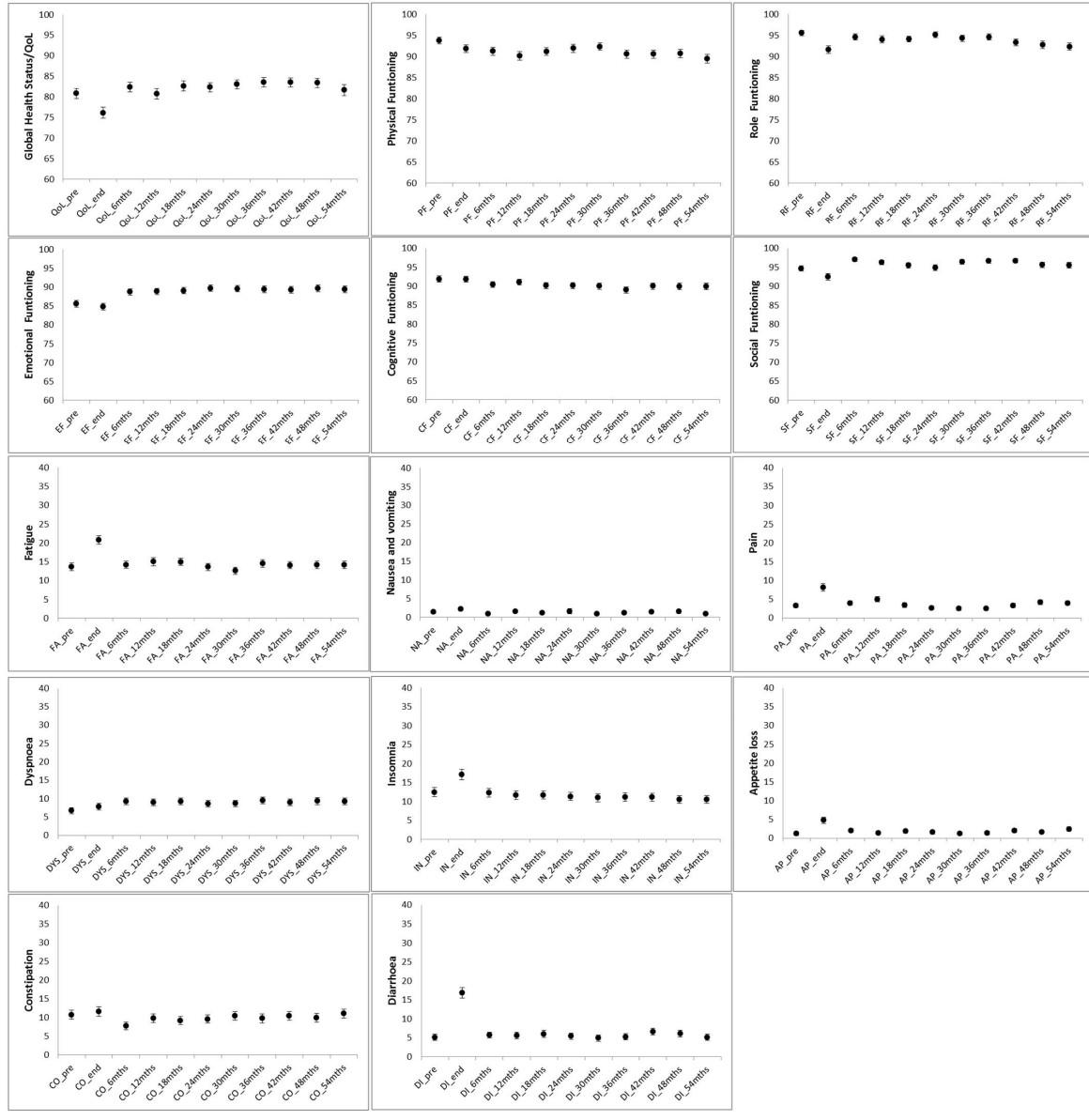

**Figure 2.** Longitudinal evaluation (5-year time frame of the EORTC QLQ C30 investigated QoL dimensions.

*3.2. Predictors of QoL Changes at RT End*

Considering the most impaired time point of the ANOVA, detailed analyses focused on the possible predictors' deterioration of HRQoL at RT end were carried out. It emerged that acute Grade $\geq 2$ GI toxicity significantly impaired global health, physical, role functioning, fatigue, appetite loss, diarrhoea, and pain ($p$-value range: 0.02–0.0003, worsening range: 3–9 points). Prophylactic lymph-nodal irradiation resulted in significantly lower QoL levels for global health, fatigue, appetite loss, and diarrhoea ($p$-value range: 0.05–0.0001, worsening range: 5–14 points). Acute Grade $\geq 2$ GU toxicity led to lower levels of role functioning and higher pain ($p = 0.03$ and 0.002, respectively, worsening 5 and 10 points, respectively). The radiation doses were associated with diarrhoea (most informative cut-off at 81 2Gy-equivalent, $p = 0.0001$, 23.9 vs. 13 points). The use of any neoadjuvant/concomitant hormonal therapy was associated with lower pain (6.7 vs. 11, $p = 0.01$), while the use of a peripheral anti-androgen (e.g., bicalutamide) was associated with lower fatigue (19.2 vs. 24.8, $p = 0.01$).

## 4. Discussion

Intense research is actively performed not only to improve prostate cancer patient clinical outcomes but also to better understand the possible impact of side effects on an individual's functioning, both physical and psychosocial. These issues may clearly have a strong impact on the patient's decision-making process. In fact, all these items not only reflect the patient's overall physical health but also are related to the ability to perform tasks associated with daily life activities and employment. In this study, we therefore investigated the longitudinal QoL changes from radiotherapy start to 5 years after IMRT-IGRT delivered for PCa within a large multicentric study. The prospective evaluation of HRQoL was performed using patient-reported QoL evaluated by means of a validated questionnaire, the EORTC QLQ-C30. In addition, the possible impact of clinical, technical, and dosimetric data on QoL were investigated, focusing on the timing corresponding to the evidence of a significant impact of the treatment on QoL.

The results of this study highlighted a relatively temporary worsening of 9 QoL dimensions out of 14 at RT end with respect to baseline, with a complete restoration within 6 months, with the sole exception of physical functioning, exhibiting an additional worsening of 4 points at the 5-year follow-up. The disappearance of the detrimental effect in certain aspects of QoL after six months can be explained by the fact that supporting nutritional status, a possible mental health serenity connected to anxiety reduction, suddenly managing symptoms during and following treatment, and good treatment tolerability improve QOL. Moreover, acute symptoms usually resolve within a few months [16] and patients present a fatigue reduction related to the end of transportation to the radiotherapy centre.

In particular, a worsening of 5 points for global health, role functioning, appetite loss, and pain and 4 points for physical functioning were detected at RT end. These findings are consistent with previous reports describing lower HRQoL and functional status following RT [4,11,13,17–19]. On the other hand, some studies reported no significant changes in daily activities during the treatment course [20,21].

The two dimensions mostly impaired by irradiation were fatigue and diarrhoea, which exhibited a worsening of 7 and 14 points, respectively, at radiotherapy end. The more likely sources of fatigue during radiotherapy may be hormonal therapy, transportation to the institute where radiotherapy is delivered, and the treatment itself, while diarrhoea represents the most common radiation side effect related to pelvic radiotherapy. Fatigue is very commonly reported in men treated with radiotherapy for prostate cancer [17,22]. Physical functioning worsening may probably be related to patients' age increasing between baseline and 60 months follow-up. In the series of Sveistrup et al. [23], a decrease in physical functioning one year after RT was observed even though it was usually small (<5 points). The authors' hypothesis was that a decrease in physical functioning lower than 5 points had to be considered clinically not significant. No variation in cognitive functioning, insomnia, nausea, dyspnoea, and constipation was observed in our series. The lack of

change in cognitive function following radiotherapy was also reported by Bansal et al. [24]. Other studies showed, on the contrary, moderate but transient impaired QoL immediately after radiotherapy [16,25]. The exact reason for the correlation of a lower degree of pain and hormonal therapy is not easily explainable. A hypothesis could be that the result is influenced by the concept called benefit finding. Namely, benefit finding refers to the fact that the beginning of pharmacological therapy could reduce patient anxiety related to the patient's illness. In fact, it was reported that symptom complications such as fatigue, sleeplessness, pain, and diarrhoea were significantly associated with levels of anxiety [26].

When focusing on QoL dimensions' variation at radiotherapy end as compared to baseline, acute Grade $\geq 2$ GI toxicity was found to significantly affect global health, physical, role functioning, fatigue, appetite loss, diarrhoea, and pain, while acute Grade $\geq 2$ GU toxicity produced lower role functioning and higher pain. Also, Sveistrup and coworkers [23], analysing the impact of urinary and gastrointestinal bothers, concluded that the worsening of both GU and GI symptoms were associated with a QoL decrease in several scales. Clark et al. [27], investigating the impact of pelvic symptoms on HRQoL scores, reached similar conclusions. Conversely, Jereczek-Fossa et al. [28], considering a cohort of 337 patients followed for 19 months after irradiation, reported no change in urinary symptom-related QoL in PCa patients treated with image-guided radiation therapy.

Prophylactic lymph-nodal irradiation resulted in significantly lower QoL for global health, fatigue, appetite loss, and diarrhoea (*p* range: 0.05–0.0001, worsening range: 5–14 points), apparently as a result of the larger volumes irradiated. Higher prescription doses were associated with an increased risk of diarrhoea, even though it was only if delivered at radiation doses exceeding 81 EQD2 Gy, possibly as a result of the more refined dose delivery achievable with modern IMRT. The impact of radiation doses on short-term bowel dysfunction, such as diarrhoea, urgency, or rectal pain, is largely reported. Acute symptoms usually resolve within a few months [16].

The use of any neoadjuvant/concomitant hormonal therapy was associated with lower pain (6.7 vs. 11, *p* = 0.01), while the use of an anti-androgen was associated with lower levels of fatigue in the range of 5 points. Sanda et al. [29] highlighted how vitality may be lower in patients who receive androgen deprivation therapy (ADT). Similarly, Langston et al. [22] and Lilleby et al. [17] reported that fatigue is a common side effect in men affected by prostate cancer, especially if receiving ADT. The direct impact of hormonal therapy on fatigue and on treatment-related symptoms was also found by Marchand et al. [30]. It is noteworthy to observe that, differently from what was observed by Krahn MD et al. [18], no impact of ADT on social functioning and global health emerged in our study.

The strength of our series is that all data were collected prospectively, including all the baseline HRQoL data collected prior to the treatment start. Moreover, the 5-year longitudinal nature of study has the potential to adequately capture even slight changes in patients' HRQoL over a robust time span. At the same time, considering the time interval of data collection in the study, it was possible to correlate the changes to specific causes. Similar prior studies mostly reported results at 2 years [31]. Thereafter, our results not only confirm the 6-month prospective published literature data on HRQoL after IMRT [19,32] but also provide additional information of HRQoL trends up to 60 months.

The fact that the patients recovered their initial QoL within 6 months is an extremely important result indicating a good tolerability of the treatment. Therefore, considering that acute toxicity is predictive of late toxicity for general toxicity as well as urinary and bowel toxicity, we will also expect less late toxicity. Moreover, the "acute" effects that disappear after six months could be in favour of hypofractionation that allows for the shortening of the radiotherapy course leading to direct patient convenience, cost savings, and potential radiobiological advantages. In fact, the literature reported that prostate cancer has a relatively low alpha–beta ratio compared to other malignancies, and even in relation to adjacent normal tissues (e.g., rectum and bladder). This suggests an increase in the therapeutic ratio with larger doses per fraction; that is, prostate cancer cells are more sensitive to hypofractionation than the surrounding organs at risk.

In this study, however, we did not have the opportunity to investigate if hormonal therapy for more or less than six months has an impact on QoL and if outcome variation during the patients' follow-up modifies the QoL score.

## 5. Conclusions

Our findings suggest that modern radiotherapy delivered by conventional or hypofractionated regimens with EQD2 doses up to 90 Gy represents a modality that does not significantly affect long-term QoL. A temporary deterioration of some investigated endpoints was experienced by patients at the end of radiotherapy, but all radiation-induced detrimental effects disappeared after six months from radiotherapy end. This result is extremely important because it indicates that there is no need for additional home health or spousal support.

We believe that nowadays the prostate cancer treatment aim is not only to prolong life, increasing tumour control rate and survival, but also to apply any effort aiming to improve QoL. Moreover, a regular QoL measurement of patients during the treatment course may be a useful instrument in order to detect QoL variation early using a personalized approach.

On the other hand, further analyses should focus on better depicting specific subgroups of patients who may be more subject to long-term impairment of HRQoL.

**Author Contributions:** Writing—original draft preparation, A.M. (Angelo Maggio), T.R., C.F. and C.C.; data curation, C.B. and F.B.; writing—review and editing, M.G., D.C., B.A., B.F., P.F., T.G., G.G., V.L., A.M. (Alessandro Magli), E.M., E.P., P.S., G.S., E.V., J.M.W., A.G. and R.V. All authors have read and agreed to the published version of the manuscript.

**Funding:** The research leading to these results has received funding from AIRC under IG 2014-ID. 14603 project—P.I. Cozzarini Cesare.

**Institutional Review Board Statement:** The study was conducted in accordance with the Declaration of Helsinki, and approved by the Ethics Committee of Fondazione Centro San Raffaele del Monte Tobar Istituto Scientifico Ospedale San Raffaele—Milano (protocol code: DUE-01 and date of approval: 4-3-2010)" for studies involving humans.

**Informed Consent Statement:** Informed consent was obtained from all subjects involved in the study.

**Data Availability Statement:** The data presented in this article is not readily available due to privacy and ethical restrictions and because are part of an ongoing study.

**Conflicts of Interest:** The authors declare no conflict of interest.

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
