# Peer review of "Quality of Life Longitudinal Evaluation in Prostate Cancer Patients from Radiotherapy Start to 5 Years after IMRT-IGRT"

_curroncol, doi:10.3390/curroncol31020062_

Round 1

Reviewer 1 Report

Comments and Suggestions for Authors

In the article "Quality of life longitudinal evaluation in prostate cancer patients from radiotherapy start to 5 years after IMRT-IGRT" Maggio et al. proposes for the first time an analysis for a longer period (60 months after treatment) that evaluates the evolution of quality of life (QoL) following IMRT for prostate cancer (PCa) and to determine possible association with clinical/treatment data. The result is interesting and the disappearance of the detrimental effect in certain aspects of QoL after 6 months should be commented. For a non-radiation oncologist, that "EQD2" could be difficult to digest. I would suggest a comparison as to the biological effect of the schemes through the persoectva of the LQ model and some prostate-specific comments regarding the alpha/beta ratio and BED related to toxicity. A logical reasoning based on this model and the "acute" effects that disappear after 6 months could argue for hypofractionation as a future perspective. Except for ADT and irradiation of large volumes, could you identify other predictors of diarrhea, fatigue, loss of appetite among the characteristics mentioned in the group of patients? A comment might be helpful.

Author Response

In the article "Quality of life longitudinal evaluation in prostate cancer patients from radiotherapy start to 5 years after IMRT-IGRT" Maggio et al. proposes for the first time an analysis for a longer period (60 months after treatment) that evaluates the evolution of quality of life (QoL) following IMRT for prostate cancer (PCa) and to determine possible association with clinical/treatment data. The result is interesting and the disappearance of the detrimental effect in certain aspects of QoL after 6 months should be commented.

We are grateful to the Reviewer for the positive feedback.  This result has been commented in the paper. Our hypothesis was that QoL restoration can be explained by  the fact that acute symptoms usually resolve within a few months, while during the treatment the radiation-induced symptoms were promptly  managed and that after radiotherapy conclusion  a fatigue reduction related to the end of patient transportation to the radiotherapy centres could be observed. 

For a non-radiation oncologist, that "EQD2" could be difficult to digest. I would suggest a comparison as to the biological effect of the schemes through the persoectva of the LQ model and some prostate-specific comments regarding the alpha/beta ratio and BED related to toxicity. A logical reasoning based on this model and the "acute" effects that disappear after 6 months could argue for hypofractionation as a future perspective.

We are grateful to the Reviewer to the appreciate suggestion. The radiobiological model applied was added and the disappearance of symptoms as support to hypofrationation  was commented.

Except for ADT and irradiation of large volumes, could you identify other predictors of diarrhea, fatigue, loss of appetite among the characteristics mentioned in the group of patients? A comment might be helpful.

All the covariates reported in Table 2 were used in order to investigate possible predictors of the considered endpoints, and . any statistically significant correlation was reported. For example were used age, BAT, neoadjuvant or analog hormonal therapy, dose, hypofractionation, acute toxicity more than Grade ≥2 according to CTCAE, smoke, diabetes, ptv prescription dose, presence of acute urinary symptoms defined as the IPSS increase at RT end of 10 or 15 points in patients with IPSS pre-RT IPSS less than 20 points. Unfortunately no correlation with blocked hormonal therapy was possible for the lack of data hormonal therapy ending.

Reviewer 2 Report

Comments and Suggestions for Authors

In the structure of the article, it is noteworthy that only 46 patients had a T3-T4 and 186+40 had a Gleason index of 7 or higher, and the number of patients with lymphadenopathy is not referenced. However, 58% of patients received androgen blockade without knowing what its duration was, greater or less than 6 months. It is important to know what the duration of the hormone blockade has been because it could correlate with the improvement in QOL. It is important to know how many patients had the pelvis irradiated. 

The QLQ-C30 test is not designed to assess cognitive impairment, recent memory, etc., there are specific tests.

Author Response

In the structure of the article, it is noteworthy that only 46 patients had a T3-T4 and 186+40 had a Gleason index of 7 or higher, and the number of patients with lymphadenopathy is not referenced. However, 58% of patients received androgen blockade without knowing what its duration was, greater or less than 6 months. It is important to know what the duration of the hormone blockade has been because it could correlate with the improvement in QOL. It is important to know how many patients had the pelvis irradiated. 

The QLQ-C30 test wa not designed to assess cognitive impairment, recent memory, etc., there are specific tests.

Thank you for the suggestions. Lymph nodal status  was added to  Table 2.

Regarding blockade hormonal therapy, in the database the end of hormonal therapy is not indicated. Therefore we were not able to recover information relative to androgen deprivation therapy length. Using the database information the statistical analysis indicated that hormonal therapy has no impact on QoL but, unfortunately,  we were not able to dichotomize in more or less than 6 month. We acknowledged in the manuscript this study limitation.

The number of patients receiving pelvic irradiation was added in Table 2.

 We agree with the reviewer with respect to the fact that the QLQ-C30 was not designed to assess a possible detrimental impact of treatment on cognitive impairment or recent memory loss, but these aspects were out of the scope of our study.

Reviewer 3 Report

Comments and Suggestions for Authors

PROMS are important measures to evaluate different treatment modalities. The EORTS-QOL30 used by the authors is a validated  overall evaluation for prostate cancer. specific items for patients treated by radiotherapy or men receiving ADT are not evaluated in this questionnaire.

How many men completed the questionnaire at different time points?

What was extent of radiotherapy (prostate or pelvic area?)

How do you explain less pain in patients receiving ADT

How many men progressed during follow up and were these still followed in this study?

Comments on the Quality of English Language

English can be improved

Author Response

PROMS are important measures to evaluate different treatment modalities. The EORTS-QOL30 used by the authors is a validated  overall evaluation for prostate cancer. Specific items for patients treated by radiotherapy or men receiving ADT are not evaluated in this questionnaire.

How many men completed the questionnaire at different time points?

Thank you for the suggestion. A Table indicating the number of patients that completed the questionnaire has been added to the paper.

What was extent of radiotherapy (prostate or pelvic area?)

The fraction of men who received prophylactic nodal  irradiation was added in Table 2.

How do you explain less pain in patients receiving ADT?

Thank you for the question. It is not easy to explain the result but a possible hypothesis has been introduced in the paper.

“An hypothesis could be that the result is influenced by the concept called benefit finding. Namely, benefit finding refers to the fact that the beginning of pharmacological therapy could reduce patient anxiety related to the patient illness. In fact, it was reported that symptom complications such as fatigue, sleeplessness, pain, and diarrhoea were significantly associated levels of anxiety”

How many men progressed during follow up and were these still followed in this study?

This is an interesting point. In fact we have started to collect follow-up data. The oncological outcome investigation was not among the initial investigation end-points. However it will surely be interesting to evaluate whether  the oncological outcome following radiotherapy could have a significant impact on QoL. Thank you for the appreciate suggestion.

Comments on the Quality of English Language

English can be improved

The English was reviewed by a native English speaker.

Round 2

Reviewer 2 Report

Comments and Suggestions for Authors

The addition of the suggested information enriches the article, it is interesting to highlight the improvement in QoL of these patients after 6 months compared to the sequelae of other treatments that may be more durable.